# Unhealthy Eating Habits and Determinants of Diet Quality in Primary Healthcare Professionals in Poland: A Cross-Sectional Study

**DOI:** 10.3390/nu16193367

**Published:** 2024-10-03

**Authors:** Małgorzata Znyk, Dorota Kaleta

**Affiliations:** Department of Hygiene and Epidemiology, Faculty of Health Sciences, Medical University of Lodz, Żeligowskiego 7/9, 90-647 Lodz, Poland; dorota.kaleta@umed.lodz.pl

**Keywords:** unhealthy eating habits, DQS, GP doctor, nurse, primary healthcare

## Abstract

Background/Objectives: The aim of this work was to understand the factors influencing the prevalence of dietary behaviors, as well as determinants of unhealthy eating and diet quality among primary care physicians and nurses in Poland. Methods: A cross-sectional study involving 161 doctors and 331 nurses was conducted in the years 2020–2022 in primary healthcare settings. Results: Unhealthy eating habits affected GPs aged 40–54 years (42.9%), females (64.3%), those of normal body weight (67.9%), representing private medical practice (67.9%), who had over 20 years of work experience (42.8%). Similarly, among the group of nurses, unhealthy eating habits were reported in subjects aged 40–54 (46.5%), individuals with normal body weights (49.5%), those with one chronic disease (38.4%), representing public medical practice (63.6%), with over 20 years of work experience (40.4%), seeing ≤100 patients during the work week (84.8%). The univariable logistic regression analyses for unhealthy dietary habits showed that overweight GPs had lower odds of unhealthy eating habits (OR = 0.35; 95% CI: 0.11–1.08; *p* < 0.05). Among the nurses, the odds of unhealthy eating habits increased with the number of years of work. Nurses with 10–20 years of work experience had 1.23 times greater odds of unhealthy eating habits, while people working for more than 20 years had 1.81 times greater odds of unhealthy eating habits than individuals working for a period shorter than ten years (OR = 1.23; 95% CI: 0.68–2.23; *p* > 0.05 vs. OR = 1.81; 95% CI: *p* < 0.05). The multivariable logistic regression analysis did not show statistically significant results. Conclusions: These issues should be addressed when planning educational activities aimed at supporting healthcare professionals in implementing lifestyle changes.

## 1. Introduction

As shown by numerous studies, the health of medical professionals, i.e., doctors and nurses, can influence the health of a larger population. Additionally, there is a link between physicians’ health practices and patient interactions [1,2]. Nurses and doctors do everything in their power to provide their patients with exceptional care, often at the detriment of their own self-care. As a result, their participation in healthy lifestyle behaviors is not prioritized [3,4].

The lifestyle of physicians affects not only their health but also doctor–patient consultations, as well as the patient’s care [1,2]. Health behaviors displayed by physicians are a strong predictor of the healthy lifestyle advice they provide [1]. It has been observed that doctors with unfavorable lifestyle habits are less proactive in advising patients than those following healthy practices [1]. The health behaviors that physicians attempt to model and the effects of these behaviors may influence their counseling practices and self-confidence in providing advice [1]. Similarly, nurses indicate that their nutritional behaviors may have an impact on the development of health and nutritional attitudes in patients [5].

It is important to remember that the main behavioral factor causing risk to population health is poor dietary habits [6]. In their professional work, healthcare specialists are often required to provide a model of healthy eating habits [6,7]. Despite the widely available knowledge about the role of healthy nutrition, irregular eating is a problem which occurs both among nurses [5,8] and doctors [9].

Healthcare workers, including doctors and nurses working in primary healthcare settings, are a professional group that performs shift work, which is associated with unhealthy lifestyle behaviors, such as poor diet and inadequate sleep hygiene [10]. Research studies show that nurses working shifts display poor health behaviors [11]. This is due to the fact that the way they choose and buy food or prepare meals is different from that of people working normal day shifts [12]. Their varied working schedule means that nurses do not prepare nutritious meals every day and do not go shopping systematically [8]. Shift workers are half as likely to eat regularly as day workers [5]. They often eat ready-to-eat, processed meals with high fat and energy content, and products with low nutritional value [13]. Additionally, they use box diets and catered meals more frequently [8]. In another study, respondents with longer shift hours reported shorter sleep times; however, they also had higher sleep quality, healthy eating habits, and lower perceived stress [6].

In the case of night shift workers, it was found that they consumed fewer vegetables and fruits [14,15], but a larger amount of fried foods compared to afternoon and morning shift workers [16]. Research conducted among residents showed a correlation between low fruit consumption and an increase in the number of overtime hours [17].

Family doctors are perceived by patients as the most reliable source of knowledge about nutrition. Therefore, general and specific recommendations have been developed for physicians to support their daily counseling practices with patients [18]. Recommendations include promoting a balanced diet, i.e., eating plenty of fruit, vegetables, whole grains and fatty fish regularly, limiting saturated fatty acids (including trans fatty acids), consuming moderate amounts of alcohol, and reducing sodium intake [18].

The scientific community is very interested in dietary patterns based on the Mediterranean diet, which has been shown to reduce the risk of stroke and coronary heart disease (CHD) [19]. A study Fung T., et al. found that the Western diet, which is based on high intake of refined grains, processed meats, and French fries, was independently associated with a higher risk of CHD. Eating a prudent diet rich in vegetables, fruit, poultry, legumes, whole grains, and fish was associated with a lower risk of CHD [19].

There is little information about the health of doctors and nurses and their lifestyles [1]. The priority of public health should be healthy medical staff, i.e., nurses [20], and doctors. Numerous studies conducted on nurses have shown a high prevalence of obesity and overweight among this group of professionals [21]. Other studies have focused on physical activity levels or diet quality among nurses [20]. Also, there have been many reports on a high prevalence of overweight and obesity related to unfavorable lifestyle habits in the medical community [1].

However, there is a lack of research studies on primary care doctors and nurses that compare poor eating habits and the quality of their diet. Our study is one of the first to address this topic in Poland. Furthermore, our article is a continuation of a previously published paper on the dietary behavior of primary care patients [22]. The aim of this work was to understand the factors influencing the prevalence of dietary behaviors, as well as determinants of unhealthy eating and diet quality among primary care physicians and nurses in Poland.

## 2. Materials and Methods

### 2.1. Characteristics of the Study Participants

A cross-sectional study among medical workers (doctors and nurses) was conducted in the years 2020–2022. The first stage (2020–2021) included family doctors working in primary healthcare and the second stage (2022) involved primary healthcare nurses. At the end of 2020, there were 211 primary healthcare entities providing services to adults in the city of Łódź [23]. A number generator was used to draw 120 numbers [24]. From a list of 211 primary healthcare entities, 100 primary healthcare entities were selected based on the first 100 randomly selected numbers. A detailed description of the methodology can be found elsewhere [2,25]. The managers of the facilities agreed to the medical workers’ participation in the study. In each of the randomly selected facilities, an interview was conducted with every second general practitioner (GP) and every second nurse. If any discrepancies occurred, a third nurse/GP was selected. Interviews with doctors were conducted in the morning on Monday or Wednesday, and in the afternoon on Tuesday or Friday. Interviews with nurses were conducted on Monday and Tuesday mornings and on Wednesday and Friday afternoons.

Based on the inclusion criterion, the study recruited family doctors/nurses providing services to adult patients (aged at least 18 years) in primary healthcare, who gave their written informed consent to participate in the study. The respondents were assured by the researcher that the survey was anonymous and that the results would be used for scientific purposes only. The respondents had the opportunity to withdraw from the study at any time.

The researcher invited 200 primary care physicians and 350 nurses to participate in the study. Ultimately, the study included 161 doctors and 331 primary care nurses working in Łódź, whereas 39 doctors and 19 nurses refused to participate in it. Doctors and nurses mentioned a large number of patients per working day and lack of time as the reason for their refusal to participate in the study. The participation rate was 80% for doctors and 95% for nurses.

The study was conducted in compliance with the ethical principles of the Declaration of Helsinki. The research project received a positive opinion from the Ethics Committee of the Medical University of Lodz (number RNN/315/18/KE, 18 September 2018).

### 2.2. Questionnaire

The research tool was an anonymous paper questionnaire. The questionnaire used for the study has been validated in other studies [26,27]. Two different questionnaires were developed for doctors and nurses, respectively. They included questions about sociodemographic characteristics (age, gender, medical practice, years of work, number of patient visits during the routine work week, number of jobs), health status (body mass index, number of chronic diseases), and health behaviors (tobacco smoking, e-cigarette use, alcohol consumption, physical activity, and number of meals consumed per day). The respondents were asked to self-assess their height, body weight and physical activity. Based on the reported height and weight, a BMI was calculated for each GP and nurse according to the following formula: body weight (kg) divided by the square of height (m^2^). Based on the calculated BMI, the respondents were divided into groups: group I with normal body weight (BMI < 25 kg/m^2^), group II including overweight individuals (BMI ≥ 25 to <30 kg/m^2^), and group three including those with obesity (BMI ≥ 30 kg/m^2^) [28].

Primary care physicians and nurses were physically active if, according to their self-report, they performed 75–150 min of vigorous physical activity or 150–300 min of moderate physical activity per week [29].

The respondents were asked to provide information on chronic diseases they had, such as type 2 diabetes, hypertension, chronic obstructive pulmonary disease, coronary heart disease, asthma, and other conditions. Based on the answers given, they were divided into four groups, comprising subjects with no diseases (group I), those with one disease (group II), two diseases (group III), and three or more diseases (group IV).

The respondents were also asked a yes–no question about smoking and the number of cigarettes they smoked per day, if any. Doctors and nurses who smoked five or more cigarettes a day were regarded as smokers. The study participants were also asked about the use of e-cigarettes and could choose the following answers: I. yes, every day, II. yes, occasionally, or III. I don’t use them at all. Those who used e-cigarettes every day were considered to be regular users. Doctors and nurses were also asked a yes–no question about their alcohol consumption over the previous seven days. Additionally, primary care workers were surveyed with regard to their type of medical practice (private, public), length of service (less than 10 years; 10–20 years; or more than 20 years of work), the number of patient visits they had during a typical working week (≤100 patients or >100 patients), and their number of jobs (one job or two, or more jobs).

Some of the questions also concerned providing advice on diet and physical activity and measuring BMI in their patients (yes–no). One of the sections of the questionnaire used for this article concerned nutrition and included questions about the frequency of consumption of fruit, vegetables, fish, and fat. The study assessed the quality of the diet and eating behaviors of primary care physicians and nurses. A detailed description of the dietary questionnaire and the validated Dietary Quality Score (DQS) used has been discussed elsewhere [22,26]. Based on this tool, it is possible to determine the type of diet (healthy, unhealthy, average) the surveyed population follows, as well as their level of consumption of specific dietary elements.

Points were assigned to dietary components (such as vegetables, fruits, fish, and fats) (Appendix A).

The participants were awarded two points for eating more than five servings of vegetables per week, one point for eating between two and five servings per week, and zero points for eating fewer than two servings per week. Eating more than three servings of fruit a day corresponded to two points, three servings a week but no fewer than two a day corresponded to one point, and fewer than three servings a week to zero points. Fat rating was divided into spreadable fats and cooking fats. In the case of spreads, two points were awarded for no fat, one point for vegetable margarine, and zero points for butter, blended spread, and lard. Fat points were summed up (Appendix A).

Based on the obtained scores, the diet followed by medical workers was determined. The more points a respondent received, the more often they ate fish, vegetables, and fruit, and the less often they used fat for spreading or cooking. The higher the number of points obtained, the better the respondents’ eating habits and the healthier their diet.

Based on the score obtained, it was determined what diet the medical workers followed. An unhealthy diet (with a total score of 0–3) indicated low consumption of vegetables, fruit, and fish. The average diet (4–6 points in total) was associated with more frequent use of fats for spreading and cooking and less frequent consumption of vegetables, fruit, and fish. A healthy diet (earning a total of 7–8 points) involved less frequent use of fats for spreading or cooking and more frequent consumption of vegetables, fruit, and fish (Appendix A).

### 2.3. Statistical Analysis

The distribution of the studied variables and descriptive statistics was presented using numbers and percentages. The population of surveyed healthcare workers (doctors, nurses) was divided into three groups: (1) unhealthy dietary habits; (2) Average Dietary Habits; (3) Healthy Dietary Habits. The Mantel–Haenszel chi-square test was used for statistical analysis. To identify the determinants of unhealthy eating behaviors among doctors and nurses, logistic regression analysis (univariate and multivariate) was performed. Odds ratios and 95% confidence intervals were calculated for each unhealthy eating behavior variable. Statistically significant variables in the univariable logistic regression analyses (*p* < 0.05) were included in the multivariable logistic regression analyses. Additionally, binary logistic regression was used to predict the presence of overweight/obesity based on dietary habits. The dependent variable (BMI group) was dichotomous (overweight/obese; not overweight/obese).

The statistical analysis was performed using the STATISTICA program (version 13.3) licensed by the Medical University of Lodz. The statistical analysis performed was a continuation of the previous one, conducted among primary care patients and published in another article [22].

## 3. Results

### 3.1. Characteristics of the Study Population

Among the 161 primary care physicians, 73.9% were women. The largest group comprised doctors aged 40–54 (38.5%). Among the study participants, 30.4% of the doctors were overweight and 11.8% were obese. Approximately 36.6% of the physicians did not report any chronic disease. More than half of them (64.6%) were employees of the private healthcare sector. Almost half of them (46.6%) had worked in their profession for over 20 years. A majority of the primary care physicians (66.5%) indicated that this was their only place of work. Only 8.7% of the doctors reported smoking cigarettes, 15.5% used e-cigarettes, and 42.2% engaged in alcohol consumption. As many as 78.3% of the doctors indicated undertaking physical activity. Three-quarters of the surveyed doctors provided advice on nutrition and physical activity to their patients and measured their body weight. Also, 65.8% of the doctors favored eating four to five meals a day (Table 1).

All the 331 primary care nurses were female. The largest group comprised nurses aged 55 and over. Among them, 38.1% were overweight and 16.9% were obese. Approximately 33.8% of the respondents had a history of one chronic disease. More than half of them (64.0%) worked in public primary healthcare. About 81.9% of the nurses reported that this was their only place of work. Smoking cigarettes was indicated by 18.1% of the nurses, use of e-cigarettes was reported by 6%, and alcohol consumption was reported by 41.4%. Only 45.6% of the women reported engaging in physical activity. Approximately 88.5% of the nurses provided counseling on physical activity and nutrition to patients and measured their body weight. Most nurses ate four to five meals a day (66.2% of the respondents) (Table 1).

### 3.2. Dietary Quality Score (DQS) among GP Doctors and Nurses

More than half (57.3%) of the medical workers consumed fish (<200 g/week; 57.8% of the GPs and 57.1% of the nurses. About three-quarters of the study population consumed two to five servings of vegetables per week, and the frequency of consumption was higher among the nurses (80.7%) than the doctors (56.5%). Among the respondents, 27.2% consumed more than five servings of vegetables per week, with the percentage rates of 43.5% and 19.3% for the doctors than nurses, respectively.

A majority of the medical workers (64.4%) consumed fruit (3 portions per week and <2 portions/day, with the frequency of consumption being comparable for the GPs (65.8%) and the nurses (63.7%). More than three portions of fruit per week were consumed by 26.2% of the respondents, more commonly by the nurses than the doctors (27.5% vs. 23.6%).

Over 70.7% of the study subjects ate fats for spreading, such as butter, blended spread, lard, and these fats were more frequently used by the nurses (75.8%) than the doctors (60.2%). Total cooking fats, such as vegetable margarine and oil, were used by 56.9% of the study population, both by the doctors and the nurses (54.7% and 58%, respectively). There were no statistically significant results (*p* > 0.05) (Table 2).

Table 3, Table 4 and Table 5 shows DQS categories and individual characteristics of primary care workers (total, GPs, and nurses). Unhealthy eating habits were practiced by 25.8% of the surveyed population, including 17.4% of the GPs and 29.9% of the nurses. Only 12.2% of the study participants engaged in healthy eating habits, including 19.2% of doctors and 8.8% of nurses. Average Dietary Habits were followed by 62% of the study population (63.4% GPs and 61.3% of nurses).

Unhealthy eating habits were most frequently practiced by medical workers aged 40–54 (45.7%), females (92.1%), individuals with normal body weight (53.6%), those with one chronic disease (36.2%), those representing public medical practice (56.7%), those with over 20 years of work experience (41.0%), and those who were seeing ≤100 patients during the working week (77.1%) (Table 3). Unhealthy eating habits affected GPs aged 40–54 years (42.9%), who were female (64.3%), of normal body weight (67.9%), representing private medical practice (67.9%), and had over 20 years of work experience (42.8%) (Table 4). Similarly, among the group of nurses, unhealthy eating habits were followed by subjects aged 40–54 (46.5%), individuals with normal body weight (49.5%), those with one chronic disease (38.4%), those representing public medical practice (63.6%), those with over 20 years of work experience (40.4%), and those who were seeing ≤100 patients during the work week (84.8%) (Table 5). There were no statistically significant results (*p* > 0.05).

Healthy eating habits were most frequently reported in medical workers aged 40–54 (41.7%; *p* < 0.05), who were female (83.3%; *p* < 0.05), with normal body weight 51.7%, representing private medical practice (53.3%), who had been working in the profession for less than ten years (38.3%) and saw ≤100 patients during the work week (60.0%) (Table 3). Among the doctors, healthy eating habits were more frequently reported in those aged 40–54 years (45.2%), females (67.7%), with normal body weight (54.8%), without chronic diseases (41.9%), representing private medical practice (77.4%) and seeing >100 patients during the work week (67.7%) (Table 4). Similarly, among the nurses, healthy eating habits were more prevalent in women with normal body weight (48.3%), without chronic diseases (37.9%), and representing public medical practice (72.4%), who had been working in the profession for less than ten years (44.8%), and who reported seeing ≤100 patients during the working week (89.7%) (Table 5). There were no statistically significant results (*p* > 0.05). Other individual characteristics of the respondents depending on their eating habits are presented in Table 3, Table 4 and Table 5.

The univariable logistic regression analyses for unhealthy dietary habits showed that overweight medical workers had lower odds of unhealthy eating habits (OR = 0.66; 95% CI: 0.42–1.05; *p* < 0.05) than professionals with normal weight and obese people. Additionally, primary care workers seeing ≤100 patients during their routine work week had higher odds of unhealthy eating habits (OR = 1.59; 95% CI: 0.99–2.55; *p* < 0.05) than those seeing >100 patients. Cigarette smokers were more than one and a half times more likely to have unhealthy eating habits (OR = 1.58; 95% CI: 0.93–2.69; *p* < 0.05) than nonsmokers. Primary care workers eating four to five meals a day had lower odds of unhealthy eating habits (OR = 0.46; 95% CI 0.20–1.03; *p* < 0.05). Overweight GPs had lower odds of unhealthy eating habits (OR = 0.35; 95% CI: 0.11–1.08; *p* < 0.05). Among the nurses, the odds of unhealthy eating habits increased with the number of years of work. Nurses with 10–20 years of work experience had a 1.23 times greater odds of unhealthy eating habits and people who had worked for more than 20 years had a 1.81 times greater odds of unhealthy eating habits than individuals who had been working for a period shorter than ten years (OR = 1.23; 95% CI: 0.68–2.23; *p* > 0.05 vs. OR = 1.81; 95% CI: *p* < 0.05). In the multivariable logistic regression, the analysis did not show statistically significant results (Appendix A).

It was found that with the increase in unhealthy eating habits within each variable defining habits, there was a noticeable rise in the odds ratio (OR) of BMI groups (overweight/obesity), and the observed regularity was statistically significant for fish consumption (<200 g/week, OR =1.27, *p* < 0.05; no intake OR = 1.31, *p* < 0.05), and fat cooking (margarine/butter/blended spread/lard, OR = 1.47, *p* < 0.05). The results are presented in Table 6.

## 4. Discussion

Our study examined the eating behaviors of primary care physicians and nurses. The determinants of an unhealthy diet have been identified.

More than half of the nurses involved in the study were overweight or obese (38.1% and 16.9%, respectively). Overweight and obesity were more common among nurses than among doctors. In the group of doctors, overweight was recorded in 30.4% and obesity in 11.8% of the subjects. A healthy medical workforce is a public health priority. Numerous studies conducted among nurses have confirmed the high prevalence of overweight and obesity in this group of medical professionals [21]. Similarly, significantly higher BMIs were reported by physicians. The study by Borgan et al. showed that three-quarters of doctors had a BMI above the normal level, and these findings mainly related to male and older doctors [1]. Obesity is also a serious public health challenge in other countries, including the United States [30]. In our study, lower rates of overweight were observed among primary care patients (25.4%) than in the population of medical workers [7]. Higher results apply to the general population in Poland (19% of obese and 39.1% of overweight individuals) [31]. Both overweight and obesity contribute to the development of many diseases that may worsen people’s quality of life. The period of the COVID-19 pandemic and changes in eating habits may have increased this risk [32].

To the best of the authors’ knowledge, this is the first study to describe the dietary patterns and eating habits of primary care staff working shifts. The obtained results emphasize that Polish doctors and nurses working shifts make unhealthy food choices.

Only 12.2% of the respondents had healthy eating habits, including 19.2% of the doctors and 8.8% of the nurses. These results were higher than in the case of primary care patients, where only 2.4% of respondents had healthy habits [22]. The popularity of healthy eating in Poland is at a similar level (15%) [33]. Other studies also confirm that education level was associated with a healthy diet. The higher the level of education, the greater the chance of practicing healthy eating habits [34,35,36].

The study showed that one-quarter of primary healthcare workers had unhealthy eating habits. Nurses (29.9%) were more likely to have unhealthy eating habits than doctors (17.4%). These results are better than those obtained for primary care patients in our other study (40%) and in other studies conducted in Poland (60%) [22,33].

The problem of a low-quality diet also affects nurses in other countries, where such dietary habits are reported by 53–61% of respondents [20,37]. Other studies have shown that the number of health-related behaviors among medical staff decreases as the number of years of work increase [38], which was also confirmed by our study.

The results of our study indicate that more than half of the surveyed medical workers, including both doctors and nurses, do not consume the recommended amount of fish (>200 g/week). Low fish consumption was also confirmed by our study conducted among primary care patients [22]. Fish consumption was also low among future doctors. About half of university students in Poland indicated that they consumed fish several times a month [39].

Every fourth medical worker in our study ate the recommended amount of vegetables and fruit. Primary care physicians more often ate the recommended amount of vegetables than nurses. In the case of fruit, they were more likely to consume the recommended number of servings.

Higher results for vegetable consumption (49.7%) and lower results for fruit consumption (13.9%) were obtained in a study that involved primary care patients [22]. In other studies, the recommended intake of fruit and vegetables refers to 44–80% of respondents [33,40]. Medical workers also often choose the wrong type of fat for spreading, with these results being slightly lower than those recorded among primary care patients (74.9%) [22]. The wrong type of fat for spreading was more frequently chosen by nurses than by doctors.

Analysis of data in Poland in 2019 (before the COVID-19 pandemic era) and 2020 (during the COVID-19 pandemic era) showed that the differences in the consumption of most product groups did not exceed 10%. A positive change was observed in the form of more frequent consumption of vegetables and fruits; only a few Polish people consumed the recommended amounts (at least twice a week). A high increase in the consumption of butter (by 14%) was observed. In the case of other animal fats and vegetable fats, no changes were noted. No differences were observed in the frequency of fish consumption [32].

In our study, a majority of the doctors and nurses declared that they consumed four to five meals, thus meeting the dietary recommendations for adults in Poland [5]. Slightly lower results were obtained in other studies, where 50% of nurses consumed four to five meals [41] or where over 60% of medical workers declared eating at least four meals [42]. In this study, 3% of the respondents (both doctors and nurses) ate one to two meals a day. The findings were better than those obtained in another study, where 8% of participants declared eating this number of meals [5]. In other studies, one-quarter of nurses left for work without eating breakfast and 30% of nurses ate one to two meals a day. Long breaks between meals can lead to an excessive drop in blood glucose levels and overeating. They also lead to increased fatigue, concentration disorders, and dizziness, which reduce work efficiency [5].

Shift work among healthcare professionals may may have an impact on eating behaviors. Polish nurses regard shift work as the reason for practicing fewer health-promoting activities [43]. Research shows that nurses perceive their professional work as one of barriers that prevent them from following a healthy diet. Day shift nurses notice more barriers to healthy eating than those working afternoon or other shifts. The lower the number of barriers perceived by nurses, the more often they follow a healthy diet [20]. Another frequently reported aspect is a lack of time to eat a proper meal at work, which affects dietary choices [11,44,45]. Lack of breaks and disturbed circadian rhythm are also mentioned as obstacles that prevent nurses from eating a healthy diet [43,45].

Research shows that doctors and nurses working shifts more often consume animal fats, processed meat, and alcohol. This may be due to the more frugal lifestyle of medical workers working in shifts and the purchase of cheaper products [5]. In Korea, nurses working shifts were more likely to overeat than nurses not working shifts [12].

Poor diet can also result from an unhealthy eating culture in the workplace, where employees often have access to snack food (cakes, donuts, cookies, e.g., during special events) [20].

Doctors and nurses should be a source of imitation not only for patients but also for themselves. About one-third of nurses believe that people at work can be role models for healthy or unhealthy eating. The nurses included in the study indicated that managers demonstrating healthy eating habits were role models for employees to follow and change their lifestyles [20].

Medical workers who eat a poor diet are more likely to suffer from chronic diseases, which cause turnover and absenteeism [46], affecting the quality of patient care [47]. It has been shown that overweight medical workers are less likely to provide counseling to overweight and obese patients than those of normal weight [48].

Preparation of healthy meals and snacks based on fruit, vegetables, nuts, cereals, vegetable oils, and legumes should be promoted among doctors and nurses. Additionally, it is recommended to eliminate processed foods, animal fats, meat, and alcohol from the diet [5]. Care should be taken to ensure that primary healthcare workers have more time for meal breaks.

Lifestyle modification should be integrated with health promotion for medical workers. A consistent schedule of shift work and rotation can have a positive impact on healthy lifestyle [6].

In our study, overweight individuals had lower odds of unhealthy eating habits compared to obese participants. It is possible that overweight subjects demonstrated lower odds of unhealthy dietary habits as a consequence of the outcome.

Medical workers who smoked were more likely to have an unhealthy diet than nonsmokers. Seeing fewer patients during the routine work week was associated with greater odds of an unhealthy diet. For nurses, more years of work were associated with greater odds of an unhealthy diet. However, this was not confirmed in multivariate logistic regression.

The other analyzed factors turned out to be insignificant determinants of choices made by primary care physicians and nurses in Poland with regard to diet quality.

The strengths of this study are as follows. Firstly, in Poland, no research study on this topic has been conducted so far among primary healthcare medical staff. This is the first study of this type that provides information on the factors influencing the prevalence of dietary habits during the COVID-19 pandemic era. Secondly, the study focused on the determinants of unhealthy eating and diet quality among primary care physicians and nurses. Thirdly, the DQS questionnaire was used to assess diet quality, and it has been proven effective in other studies.

The weaknesses of the study are as follows. Firstly, this was a cross-sectional study performed at a single time point, which means it was impossible to observe changes over longer periods. We do not know what influenced the unhealthy or healthy eating habits of medical staff. It should be emphasized that changing one’s lifestyle, including one’s eating habits, is a long-term process.

Secondly, the questionnaire did not include questions about barriers affecting unhealthy eating habits.

Thirdly, the study results may involve systematic errors underlying the investigation conducted during the COVID-19 pandemic era. During that period, health behaviors, including dietary behaviors, may have changed. Working habits and rhythms among healthcare professionals may have caused some changes (including weight changes). Also, dietary habits were assessed based on data from questionnaires completed by medical workers, which may be associated with recall bias. Similarly, weight and height information were self-assessed.

Moreover, the survey did not include a question about income, which obviously may affect food choice. Furthermore, a lack of association in multivariate analyses may result from the small sample size.

Changing unhealthy eating habits is a long-term process that needs to be verified and repeated. Primary care doctors and nurses should be role models for their patients. Therefore, it is so important that medical professionals adopt and maintain good eating habits. It is required that further research be conducted on the influence of primary healthcare staff on health behaviors among patients. Healthcare workers will be better public health educators and will motivate their patients to adopt good eating habits if they practice them themselves. The results can help governments develop appropriate strategies to support healthy lifestyles for doctors, nurses, and their patients.

## 5. Conclusions

The prevalence of unhealthy eating behaviors among primary care physicians and nurses in Poland is slightly lower than in the general population. The study indicates the need for nutritional education of doctors and nurses working shifts about regular consumption of good quality meals.

The study showed that, in the case of medical workers, the factors that significantly increased the odds of unhealthy eating behaviors were the number of patients seen during a routine working week and smoking, whereas in the case of nurses, it was the number of years of work.

These issues should be addressed when planning educational activities aimed at supporting healthcare professionals in implementing lifestyle changes.

The results also provide the basis for training medical workers in the field of a healthy diet, as well as developing health programs used by doctors and nurses in their professional work with patients.

## Figures and Tables

**Table 1 nutrients-16-03367-t001:** Characteristics of primary care medical workers (N = 492).

Variable	Total N = 492 (%)	GP Doctor n = 161 (%)	Nurse n = 331 (%)	*p*-Value
Age (years)				
<40	102 (20.7)	55 (34.2)	47 (14.2)	*p* < 0.05
40–54	200 (40.7)	62 (38.5)	138 (41.7)	
55+	190 (38.6)	44 (27.3)	146 (44.1)	
Sex				
Female	450 (91.5)	119 (73.9)	331 (100.0)	*p* < 0.05
Male	42 (8.5)	42 (26.1)	-	
Body mass index BMI				
<25 kg/m^2^	242 (49.2)	93 (57.8)	149 (45.0)	*p* < 0.05
≥25–<30 kg/m^2^	175 (35.6)	49 (30.4)	126 (38.1)	
≥30 kg/m^2^	75 (15.2)	19 (11.8)	56 (16.9)	
Number of chronic diseases				
0	160 (32.5)	59 (36.6)	101 (30.5)	*p* ≥ 0.05
1	157 (31.9)	45 (28.0)	112 (33.8)	
2	74 (15.1)	27 (16.8)	47 (14.2)	
≥3	101 (20.5)	30 (18.6)	71 (21.5)	
Medical practice				
Private	223 (45.3)	104 (64.6)	119 (36.0)	*p* < 0.05
Public	269 (54.7)	57 (35.4)	212 (64.0)	
Years of work				
<10	164 (33.3)	46 (28.6)	118 (35.7)	*p* < 0.05
10–20	145 (29.5)	40 (24.8)	105 (31.7)	
>20	183 (37.2)	75 (46.6)	108 (32.6)	
Number of patient visits during the routine working week				
≤100	346 (70.3)	59 (36.6)	287 (86.7)	*p* < 0.05
>100	146 (29.7)	102 (63.4)	44 (13.3)	
Appropriate training to provide counseling on nutrition, physical activity and weight management				
Yes	415 (84.3)	122 (75.8)	293 (88.5)	*p* < 0.05
No	77 (15.7)	39 (24.2)	38 (11.5)	
Taking measurements of body weight, height, and BMI				
Yes	350 (71.1)	125 (77.6)	225 (68.0)	*p* < 0.05
No	142 (28.9)	36 (22.4)	106 (32.0)	
Tobacco smoking				
Yes	74 (15.0)	14 (8.7)	60 (18.1)	*p* < 0.05
No	418 (85.0)	147 (91.3)	271 (81.9)	
E-cigarette use				
Yes	45 (9.1)	25 (15.5)	20 (6.0)	*p* < 0.05
No	447 (90.9)	136 (84.5)	311 (94.0)	
Alcohol consumption				
Yes	205 (41.7)	68 (42.2)	137 (41.4)	*p* ≥ 0.05
No	287 (58.3)	93 (57.8)	194 (58.6)	
Physical activity				
Yes	277 (56.3)	126 (78.3)	151 (45.6)	*p* < 0.05
No	215 (43.7)	35 (21.7)	180 (54.4)	
Number of jobs				
1	378 (76.8)	107 (66.5)	271 (81.9)	*p* < 0.05
≥2	114 (23.2)	54 (33.5)	60 (18.1)	
Number of meals during the day				
≤2	17 (3.5)	6 (3.7)	11 (3.3)	*p* ≥ 0.05
3	123 (25.0)	44 (27.3)	79 (23.9)	
4–5	325 (66.1)	106 (65.8)	219 (66.2)	
≥6	27 (5.4)	5 (3.2)	22 (6.6)	

**Table 2 nutrients-16-03367-t002:** Frequency of consumption of the most important food ingredients among primary healthcare workers (N = 492).

Food	Frequency	Total N = 492 (%)	GP Doctor n = 161 (%)	Nurse n = 331 (%)	*p*-Value
Fish	>200 g/week	83 (16.9)	30 (18.6)	53 (16.0)	*p* ≥ 0.05
	<200 g/week	282 (57.3)	93 (57.8)	189 (57.1)	*p* ≥ 0.05
	no intake	127 (25.8)	38 (23.6)	89 (26.9)	*p* ≥ 0.05
Vegetables	>5 servings/week	134 (27.2)	70 (43.5)	64 (19.3)	*p* ≥ 0.05
	2–5 servings/week	358 (72.8)	91 (56.5)	267 (80.7)	*p* ≥ 0.05
	<2 servings/week	-	-	-	-
Fruit	>3 pieces/day	129 (26.2)	38 (23.6)	91 (27.5)	*p* ≥ 0.05
	3 pieces/week and <2 pieces/day	317 (64.4)	106 (65.8)	211 (63.7)	*p* ≥ 0.05
	<3 pieces/week	46 (9.4)	17 (10.6)	29 (8.8)	*p* ≥ 0.05
Fat	none	89 (18.1)	48 (29.8)	41 (12.4)	*p* ≥ 0.05
spread	margarine, vegetable margarine	55 (11.2)	16 (10.0)	39 (11.8)	*p* ≥ 0.05
	butter, blended spread, lard	348 (70.7)	97 (60.2)	251 (75.8)	*p* ≥ 0.05
cooking	none/olive oil	36 (7.3)	18 (11.1)	18 (5.4)	*p* ≥ 0.05
	vegetable margarine, oil	280 (56.9)	88 (54.7)	192 (58.0)	*p* ≥ 0.05
	margarine/butter/blended spread/lard	176 (35.8)	55 (34.2)	121 (36.6)	*p* ≥ 0.05

**Table 3 nutrients-16-03367-t003:** DQS categories and individual characteristics of primary care workers (total).

		Unhealthy Dietary Habits		Average Dietary Habits		Healthy Dietary Habits	
	Total N = 492 (%)	n = 127 (25.8%)	*p*-Value	n = 305 (62.0%)	*p*-Value	n = 60 (12.2%)	*p*-Value
Age (years)							
<40	102 (20.7)	25 (19.7)	*p *≥ 0.05	58 (19.0)	*p *≥ 0.05	19 (31.7)	*p* < 0.05
40–54	200 (40.7)	58 (45.7)		117 (38.4)		25 (41.7)	
55+	190 (38.6)	44 (34.6)		130 (42.6)		16 (26.6)	
Sex							
Female	450 (91.5)	117 (92.1)	*p *≥ 0.05	283 (92.8)	*p *≥ 0.05	50 (83.3)	*p* < 0.05
Male	42 (8.5)	10 (7.9)		22 (7.2)		10 (16.7)	
Body mass index BMI							
<25 kg/m^2^	242 (49.2)	68 (53.6)	*p *≥ 0.05	143 (46.9)	*p* < 0.05	31 (51.7)	*p *≥ 0.05
≥25–<30 kg/m^2^	175 (35.6)	36 (28.3)		121 (39.7)		18 (30.0)	
≥30 kg/m^2^	75 (15.2)	23 (18.1)		41 (13.4)		11 (18.3)	
Number of chronic diseases							
0	160 (32.5)	37 (29.1)	*p *≥ 0.05	99 (32.5)	*p *≥ 0.05	24 (40.0)	*p *≥ 0.05
1	157 (31.9)	46 (36.2)		93 (30.5)		18 (30.0)	
2	74 (15.1)	19 (15.0)		47 (15.4)		8 (13.3)	
≥3	101 (20.5)	25 (19.7)		66 (21.6)		10 (16.7)	
Medical practice							
Private	223 (45.3)	55 (43.3)	*p *≥ 0.05	136 (44.6)	*p *≥ 0.05	32 (53.3)	*p *≥ 0.05
Public	269 (54.7)	72 (56.7)		169 (55.4)		28 (46.7)	
Years of work							
<10	164 (33.3)	37 (29.1)	*p *≥ 0.05	104 (34.1)	*p *≥ 0.05	23 (38.3)	*p *≥ 0.05
10–20	145 (29.5)	38 (29.9)		88 (28.9)		19 (31.7)	
>20	183 (37.2)	52 (41.0)		113 (37.0)		18 (30.0)	
Number of patient visits during the routine working week							
≤100	346 (70.3)	98 (77.1)	*p *≥ 0.05	212 (69.5)	*p *≥ 0.05	36 (60.0)	*p *≥ 0.05
>100	146 (29.7)	29 (22.9)		93 (30.5)		24 (40.0)	
Appropriate training to provide counseling on nutrition, physical activity, and weight management							
Yes	415 (84.3)	110 (86.6)	*p *≥ 0.05	256 (84.0)	*p *≥ 0.05	49 (81.7)	*p *≥ 0.05
No	77 (15.7)	17 (13.4)		49 (16.0)		11 (18.3)	
Taking measurements of body weight, height, and BMI							
Yes	350 (71.1)	90 (70.9)	*p *≥ 0.05	215 (70.5)	*p *≥ 0.05	45 (75.0)	*p *≥ 0.05
No	142 (28.9)	37 (29.1)		90 (29.5)		15 (25.0)	
Tobacco smoking							
Yes	74 (15.0)	25 (19.7)	*p *≥ 0.05	44 (14.4)	*p *≥ 0.05	5 (8.3)	*p *≥ 0.05
No	418 (85.0)	102 (80.3)		261 (85.6)		55 (91.7)	
E-cigarette use							
Yes	45 (9.1)	9 (7.1)	*p *≥ 0.05	31 (10.2)	*p *≥ 0.05	5 (8.3)	*p *≥ 0.05
No	447 (90.9)	118 (92.9)		274 (89.8)		55 (91.7)	
Alcohol consumption							
Yes	205 (41.7)	50 (39.4)	*p *≥ 0.05	130 (42.6)	*p *≥ 0.05	25 (41.7)	*p *≥ 0.05
No	287 (58.3)	77 (60.6)		175 (57.4)		35 (58.3)	
Physical activity							
Yes	277 (56.3)	66 (52.0)	*p *≥ 0.05	172 (56.4)	*p *≥ 0.05	39 (65.0)	*p* ≥ 0.05
No	215 (43.7)	61 (48.0)		133 (43.6)		21 (35.0)	
Number of jobs							
1	378 (76.8)	101 (79.5)	*p *≥ 0.05	232 (76.1)	*p *≥ 0.05	45 (75.0)	*p *≥ 0.05
≥2	114 (23.2)	26 (20.5)		73 (23.9)		15 (25.0)	
Number of meals during the day							
≤2	17 (3.5)	5 (3.9)	*p *≥ 0.05	10 (3.3)	*p *≥ 0.05	2 (3.3)	*p *≥ 0.05
3	123 (25.0)	33 (26.0)		81 (26.6)		9 (15.0)	
4–5	325 (66.1)	78 (61.4)		203 (66.5)		44 (73.4)	
≥6	27 (5.4)	11 (8.7)		11 (3.6)		5 (8.3)	

**Table 4 nutrients-16-03367-t004:** DQS categories and individual characteristics of primary care workers (GP doctors) (continued).

		Unhealthy Dietary Habits		Average Dietary Habits		Healthy Dietary Habits	
	GP Doctor n = 161 (%)	n = 28 (17.4%)	*p*-Value	n = 102 (63.4%)	*p*-Value	n = 31 (19.2%)	*p*-Value
Age (years)							
<40	55 (34.2)	10 (35.7)	*p* ≥ 0.05	33 (32.2)	*p *≥ 0.05	12 (38.7)	*p *≥ 0.05
40–54	62 (38.5)	12 (42.9)		36 (35.3)		14 (45.2)	
55+	44 (27.3)	6 (21.4)		33 (32.2)		5 (16.1)	
Sex							
Female	119 (73.9)	18 (64.3)	*p* ≥ 0.05	80 (78.4)	*p *≥ 0.05	21 (67.7)	*p *≥ 0.05
Male	42 (26.1)	10 (35.7)		22 (21.6)		10 (32.3)	
Body mass index BMI							
<25 kg/m^2^	93 (57.8)	19 (67.9)	*p *≥ 0.05	57 (55.9)	*p* < 0.05	17 (54.8)	*p *≥ 0.05
≥25–<30 kg/m^2^	49 (30.4)	4 (14.3)		37 (36.3)		8 (25.8)	
≥30 kg/m^2^	19 (11.8)	5 (17.8)		8 (7.8)		6 (19.4)	
Number of chronic diseases							
0	59 (36.6)	8 (28.6)	*p *≥ 0.05	38 (37.3)	*p *≥ 0.05	13 (41.9)	*p *≥ 0.05
1	45 (28.0)	8 (28.6)		28 (27.5)		9 (29.1)	
2	27 (16.8)	7 (25.0)		15 (14.7)		5 (16.1)	
≥3	30 (18.6)	5 (17.8)		21 (20.5)		4 (12.9)	
Medical practice							
Private	104 (64.6)	19 (67.9)	*p *≥ 0.05	61 (59.8)	*p *≥ 0.05	24 (77.4)	*p *≥ 0.05
Public	57 (35.4)	9 (32.1)		41 (40.2)		7 (22.6)	
Years of work							
<10	46 (28.6)	8 (28.6)	*p *≥ 0.05	28 (27.5)	*p *≥ 0.05	10 (32.3)	*p *≥ 0.05
10–20	40 (24.8)	8 (28.6)		22 (21.6)		10 (32.3)	
>20	75 (46.6)	12 (42.8)		52 (50.9)		11 (35.4)	
Number of patient visits during the routine working week							
≤100	59 (36.6)	14 (50.0)	*p *≥ 0.05	35 (34.3)	*p *≥ 0.05	10 (32.3)	*p *≥ 0.05
>100	102 (63.4)	14 (50.0)		67 (65.7)		21 (67.7)	
Appropriate training to provide counseling on nutrition, physical activity, and weight management							
Yes	122 (75.8)	23 (82.1)	*p *≥ 0.05	75 (73.5)	*p *≥ 0.05	24 (77.4)	*p *≥ 0.05
No	39 (24.2)	5 (17.9)		27 (26.5)		7 (22.6)	
Taking measurements of body weight, height, and BMI							
Yes	125 (77.6)	23 (82.1)	*p *≥ 0.05	76 (74.5)	*p *≥ 0.05	26 (83.9)	*p *≥ 0.05
No	36 (22.4)	5 (17.9)		26 (25.5)		5 (16.1)	
Tobacco smoking							
Yes	14 (8.7)	3 (10.7)	*p *≥ 0.05	10 (9.8)	*p *≥ 0.05	1 (3.2)	*p *≥ 0.05
No	147 (91.3)	25 (89.3)		92 (90.2)		30 (96.8)	
E-cigarette use							
Yes	25 (15.5)	3 (10.7)	*p *≥ 0.05	20 (19.6)	*p *≥ 0.05	2 (6.5)	*p *≥ 0.05
No	136 (84.5)	25 (89.3)		82 (80.4)		29 (93.5)	
Alcohol consumption							
Yes	68 (42.2)	11 (39.3)	*p *≥ 0.05	43 (42.2)	*p *≥ 0.05	14 (45.2)	*p *≥ 0.05
No	93 (57.8)	17 (60.7)		59 (57.8)		17 (54.8)	
Physical activity							
Yes	126 (78.3)	19 (67.9)	*p *≥ 0.05	82 (80.4)	*p *≥ 0.05	25 (80.6)	*p *≥ 0.05
No	35 (21.7)	9 (32.1)		20 (19.6)		6 (19.4)	
Number of jobs							
1	107 (66.5)	19 (67.9)	*p *≥ 0.05	68 (66.7)	*p *≥ 0.05	20 (64.5)	*p *≥ 0.05
≥2	54 (33.5)	9 (32.1)		34 (33.3)		11 (35.5)	
Number of meals during the day							
≤2	6 (3.7)	-	*p *≥ 0.05	5 (4.9)	*p *≥ 0.05	1 (3.2)	*p *≥ 0.05
3	44 (27.3)	11 (39.3)		28 (27.5)		5 (16.1)	
4–5	106 (65.8)	17 (60.7)		66 (64.7)		23 (74.2)	
≥6	5 (3.2)	-		3 (2.9)		2 (6.5)	

**Table 5 nutrients-16-03367-t005:** DQS categories and individual characteristics of primary care workers (nurses) (continued).

		Unhealthy Dietary Habits		Average Dietary Habits		Healthy Dietary Habits	
	Nurse n = 331 (%)	n = 99 (29.9%) Nurse	*p*-Value	n = 203 (61.3%)	*p*-Value	n = 29 (8.8%)	*p*-Value
Age (years)							
<40	47 (14.2)	15 (15.1)	*p *≥ 0.05	25 (12.3)	*p *≥ 0.05	7 (24.2)	*p *≥ 0.05
40–54	138 (41.7)	46 (46.5)		81 (39.9)		11 (37.9)	
55+	146 (44.1)	38 (38.4)		97 (47.8)		11 (37.9)	
Body mass index BMI							
<25 kg/m^2^	149 (45.0)	49 (49.5)	*p *≥ 0.05	86 (42.4)	*p *≥ 0.05	14 (48.3)	*p *≥ 0.05
≥25–<30 kg/m^2^	126 (38.1)	32 (32.3)		84 (41.4)		10 (34.5)	
≥30 kg/m^2^	56 (16.9)	18 (18.2)		33 (16.2)		5 (17.2)	
Number of chronic diseases							
0	101 (30.5)	29 (29.3)	*p *≥ 0.05	61 (30.0)	*p *≥ 0.05	11 (37.9)	*p *≥ 0.05
1	112 (33.8)	38 (38.4)		65 (32.0)		9 (31.1)	
2	47 (14.2)	12 (12.1)		32 (15.8)		3 (10.3)	
≥3	71 (21.5)	20 (20.2)		45 (22.2)		6 (20.7)	
Medical practice							
Private	119 (36.0)	36 (36.4)	*p *≥ 0.05	75 (36.9)	*p *≥ 0.05	8 (27.6)	*p *≥ 0.05
Public	212 (64.0)	63 (63.6)		128 (63.1)		21 (72.4)	
Years of work							
<10	118 (35.7)	29 (29.3)	*p *≥ 0.05	76 (37.4)	*p *≥ 0.05	13 (44.8)	*p *≥ 0.05
10–20	105 (31.7)	30 (30.3)		66 (32.5)		9 (31.1)	
>20	108 (32.6)	40 (40.4)		61 (30.1)		7 (24.1)	
Number of patient visits during the routine working week							
≤100	287 (86.7)	84 (84.8)	*p *≥ 0.05	177 (87.2)	*p *≥ 0.05	26 (89.7)	*p *≥ 0.05
>100	44 (13.3)	15 (15.2)		26 (12.8)		3 (10.3)	
Appropriate training to provide counseling on nutrition, physical activity, and weight management							
Yes	293 (88.5)	87 (87.9)	*p *≥ 0.05	181 (89.2)	*p *≥ 0.05	25 (86.2)	*p *≥ 0.05
No	38 (11.5)	12 (12.1)		22 (10.8)		4 (13.8)	
Making measurements of body weight, height, BMI							
Yes	225 (68.0)	67 (67.7)	*p *≥ 0.05	139 (68.5)	*p *≥ 0.05	19 (65.5)	*p *≥ 0.05
No	106 (32.0)	32 (32.3)		64 (31.5)		10 (34.5)	
Tobacco smoking							
Yes	60 (18.1)	22 (22.2)	*p *≥ 0.05	34 (16.7)	*p *≥ 0.05	4 (13.8)	*p *≥ 0.05
No	271 (81.9)	77 (77.8)		169 (83.3)		25 (86.2)	
E-cigarette use							
Yes	20 (6.0)	6 (6.1)	*p *≥ 0.05	11 (5.4)	*p *≥ 0.05	3 (10.3)	*p *≥ 0.05
No	311 (94.0)	93 (93.9)		192 (94.6)		26 (89.7)	
Alcohol consumption							
Yes	137 (41.4)	39 (39.4)	*p *≥ 0.05	87 (42.9)	*p *≥ 0.05	11 (37.9)	*p *≥ 0.05
No	194 (58.6)	60 (60.6)		116 (57.1)		18 (62.1)	
Physical activity							
Yes	151 (45.6)	47 (47.5)	*p *≥ 0.05	90 (44.3)	*p *≥ 0.05	14 (48.3)	*p *≥ 0.05
No	180 (54.4)	52 (52.5)		113 (55.7)		15 (51.7)	
Number of jobs							
1	271 (81.9)	82 (82.8)	*p *≥ 0.05	164 (80.8)	*p *≥ 0.05	25 (86.2)	*p *≥ 0.05
≥2	60 (18.1)	17 (17.2)		39 (19.2)		4 (13.8)	
Number of meals during the day							
≤2	11 (3.3)	5 (5.1)	*p *≥ 0.05	5 (2.5)	*p* < 0.05	1 (3.5)	*p *≥ 0.05
3	79 (23.9)	22 (22.2)		53 (26.1)		4 (13.8)	
4–5	219 (66.2)	61 (61.6)		137 (67.5)		21 (72.4)	
≥6	22 (6.6)	11 (11.1)		8 (3.9)		3 (10.3)	

**Table 6 nutrients-16-03367-t006:** The impact of eating behaviors on the occurrence of overweight/obesity (BMI ≥ 25 kg/m^2^)-a binary regression.

Food	Frequency	Total N = 492 (%)	OR—Odds Ratio	95% CI-Confidence Intervals	*p*-Value
Fish	>200 g/week	83 (16.9)	1.00	Ref.	Ref.
	<200 g/week	282 (57.3)	1.27	1.39–3.53	*p* < 0.05
	no intake	127 (25.8)	1.31	1.08–3.07	*p* < 0.05
Vegetables	>5 servings/week	134 (27.2)	1.00	Ref.	Ref.
	2–5 servings/week	358 (72.8)	1.22	0.65–1.43	*p *≥ 0.05
	<2 servings/week	-	-	-	-
Fruit	>3 pieces/day	129 (26.2)	1.00	Ref.	Ref.
	3 pieces/week and <2 pieces/day	317 (64.4)	1.23	0.64–1.43	*p *≥ 0.05
	<3 pieces/week	46 (9.4)	1.39	0.40–1.49	*p *≥ 0.05
Fat	none	89 (18.1)	1.00	Ref.	Ref.
spread	margarine, vegetable margarine	55 (11.2)	1.39	0.36–1.33	*p *≥ 0.05
	butter, blended spread, lard	348 (70.7)	1.27	0.57–1.45	*p *≥ 0.05
cooking	none/olive oil	36 (7.3)	1.00	Ref.	Ref.
	vegetable margarine, oil	280 (56.9)	1.45	0.24–1.06	*p *≥ 0.05
	margarine/butter/blended spread/lard	176 (35.8)	1.47	0.21–0.97	*p* < 0.05

## Data Availability

The datasets analyzed during the current study are available from the corresponding author on reasonable request.

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
