# Peer review of "Unhealthy Eating Habits and Determinants of Diet Quality in Primary Healthcare Professionals in Poland: A Cross-Sectional Study"

_nutrients, 2024, doi:10.3390/nu16193367_

Round 1

Reviewer 1 Report

Comments and Suggestions for Authors

Małgorzata Znyk and Dorota Kaleta submitted to Nutrients an article, dealing with a cross-sectional study on the unhealthy eating habits and determinants of diet quality in primary HCWs based in Lodz, Poland.

This study appears dated (2020-2022) and the outcomes that emerged may be related to a bias underlying an investigation conducted in the pandemic era, whose habits and work rhythms for HCWs may have led to variations (including weight) that cannot be ignored. Therefore, the results must certainly be updated, providing a critical reading also in this sense.

The limits of this study and the potential consequences and development hypotheses in terms of Public Health must be clearly expressed.

Comments on the Quality of English Language

Minor English editing required

Author Response

Response to Reviewer 1 Comments

We would like to thank Reviewer 1 for the positive opinion of our manuscript.

We have carefully considered all reviewer's considerations of the paper. Please find enclosed our detailed answers to reviewer queries.

Point 1:

This study appears dated (2020-2022) and the outcomes that emerged may be related to a bias underlying an investigation conducted in the pandemic era, whose habits and work rhythms for HCWs may have led to variations (including weight) that cannot be ignored. Therefore, the results must certainly be updated, providing a critical reading also in this sense.

The limits of this study and the potential consequences and development hypotheses in terms of Public Health must be clearly expressed.

Response 1:

We agree with the reviewer that the study is not the latest, but we believe that the results obtained are valuable and worth publishing. They fill research gaps in this topic in Poland. Cross-sectional studies have not been conducted among primary health care workers in Poland in recent years. They were also not conducted during the COVID-19 pandemic. Our article is a continuation of our other article "Dietary Behavior and Determinants of Diet Quality among Primary Health Care Patients in Poland" Znyk M, Raciborski F, Kaleta D., which included patients.

We have added a paragraph in the Discussion taking into account the COVID-19 pandemic. We identified the limitations of this study and potential implications and developmental hypotheses in a public health context.

Point 2:

Comments on the Quality of English Language. Minor English editing required

Response 2:

The article has been checked by an English native speaker.

Reviewer 2 Report

Comments and Suggestions for Authors

Thank you for the manuscript. Please find my comments below:

  1. I suggest that the study title specify the country in which the research was conducted: "...in primary healthcare professionals in Poland: a cross-sectional study."

  2. In the Methods section, although the study protocol has been previously published, I recommend expanding the section on participant recruitment. Specifically, include details on how participants were recruited, how written consent was obtained, and any reasons for refusal to participate.

  3. It is also important to specify the food groups included in the dietary questionnaire and provide an explanation of how the total score was calculated, as this is a key variable in your study. Additionally, please include a reference to support your classification of dietary habits as healthy, average, or unhealthy.

  4. You employ multiple covariates in this study (e.g., smoking, alcohol consumption, physical activity, BMI, and number of chronic diseases—please specify the diseases). I suggest providing a description for each covariate, including response options and details on how final scores were calculated, if applicable. Additionally, clarify whether weight, height, and physical activity were measured objectively or self-reported.

  5. In the Results section, I recommend standardizing the presentation of p-values. Use either "p < 0.05" or "p ≥ 0.05" (as in Table 1), or provide the exact p-value (as in Table 2) consistently throughout.

  6. In the title of Table 4, it is unclear what is being predicted—is it overweight/obesity? Is binary or ordinal regression used? Please clarify how the dependent variable was constructed. Additionally, ensure the presentation of p-values is consistent.

  7. Discussion, line 369: Please avoid causal language. To assess development, a prospective cohort study would be needed. This is a wrong interpretation of the cross-sectional study results. Possibly overweight subjects demonstrate lower odds of unhealthy dietary habits as a consequence of the outcome. Also, I recommend revising similar instances throughout the manuscript. You can only declare the differences in prevalence or increased odds (not risk).

  8. Conclusion, line 388: Should this refer to incidence or prevalence? Line 393: You mention correlating, but no correlations were presented in the results. Possible demonstrating an increased odds of unhealthy dietary habits? Please revise this for accuracy.

Comments on the Quality of English Language

Moderate English editing is required after revision with specific attention to epidemiological terms (incidence vs. prevalence, odds vs. risk, etc.).

Author Response

Response to Reviewer 2 Comments

We would like to thank Reviewer 2 for the positive opinion of our manuscript.

We have carefully considered all reviewer's considerations of the paper. Please find enclosed our detailed answers to reviewer queries.

Point 1:

I suggest that the study title specify the country in which the research was conducted: "...in primary healthcare professionals in Poland: a cross-sectional study."

Response 1:

We changed the title according to the reviewer's suggestion.

Point 2:

In the Methods section, although the study protocol has been previously published, I recommend expanding the section on participant recruitment. Specifically, include details on how participants were recruited, how written consent was obtained, and any reasons for refusal to participate.

Response 2:

We corrected it. We have expanded the Methods section on participant recruitment: details on how participants were recruited, how written consent was obtained, and reasons for refusing to participate. We added: ,,Based on the inclusion criterion, the study recruited family doctors/nurses providing services to adult patients (aged at least 18 years) in primary health care, who gave their written informed consent for participation in the study. The respondents were assured by the researcher that the survey was anonymous and that the results would be used for scientific purposes only. The respondents had the opportunity to withdraw from the study at any time. The researcher invited 200 primary care physicians and 350 nurses to participate in the study.(…) Doctors and nurses mentioned a large number of patients per working day and lack of time as the reason for their refusal to participate in the study.”

Point 3:

It is also important to specify the food groups included in the dietary questionnaire and provide an explanation of how the total score was calculated, as this is a key variable in your study. Additionally, please include a reference to support your classification of dietary habits as healthy, average, or unhealthy.

Response 3: We corrected it. We determined the food groups included in the dietary questionnaire and reported how the final result was calculated. We have provided a reference to classify eating habits as healthy, average, and unhealthy. We added: ,,The participants were awarded two points for eating more than five servings of vegetables per week, one point for eating between two and five servings per week, and zero points for eating fewer than two servings per week. Eating more than three servings of fruit a day corresponded to two points, three servings a week but not fewer than two a day corresponded to one point, and fewer than three servings a week to zero points. Fat rating was divided into spreadable fats and cooking fats. In the case of spreads, two paints were awarded for no fat, one point for vegetable margarine, and zero points for butter, blended spread and lard. Fat points were summed up (…). Based on the obtained scores, the diet followed by medical workers was determined. The more points a respondent received, the more often they ate fish, vegetables and fruit, and the less often they used fat for spreading or cooking. The higher the number of points obtained, the better were the respondents' eating habits and the healthier their diet.” 

Point 4:

You employ multiple covariates in this study (e.g., smoking, alcohol consumption, physical activity, BMI, and number of chronic diseases—please specify the diseases). I suggest providing a description for each covariate, including response options and details on how final scores were calculated, if applicable. Additionally, clarify whether weight, height, and physical activity were measured objectively or self-reported.

 Response 4: We corrected it. We have provided a description for each covariate, response options, and details on how to calculate the final results. Weight, height, and physical activity were measured by self-report.We added: ,, The respondents were asked to self-assess their height, body weight and physical activity. Based on the reported height and weight, BMI was calculated for each GP and nurse according to the following formula: body weight (kg) divided by the square of height (m2). Based on the calculated BMI, the respondents were divided into groups: group I with normal body weight (BMI < 25 kg/m2), group II including overweight individuals (BMI  ≥25 to <30 kg/m2) and group three including those with obesity (BMI ≥30 kg/m2). Primary care physicians and nurses were physically active if, according to their self-report, they performed 75–150 minutes of vigorous physical activity or 150–300 minutes of moderate physical activity per week.The respondents were asked to provide information on chronic diseases they had, such as type 2 diabetes, hypertension, chronic obstructive pulmonary disease, coronary heart disease, asthma, and others. Based on the answers given, they were divided into four groups, i.e. including subjects with no diseases (group I), those with one disease (group II), two diseases (group III) and three or more diseases (group IV).The respondents were also asked a yes-no question about smoking, and the number of cigarettes smoked per day, if any. Doctors and nurses who smoked five or more cigarettes a day were regarded as smokers. The study participants were also asked about the use of e-cigarettes and could choose the following answers: I. yes, every day, II. yes, occasionally, III. I don't use them at all. Those who used e-cigarettes every day were considered to be regular users. Doctors and nurses were also asked a yes-no question about alcohol consumption in the previous seven days. Additionally, primary care workers were surveyed with regard to medical practice (private, public), length of service (less than ten years; 10-20 years or more than 20 years of work), number of patient visits during a typical working week (≤ 100 patients or > 100 patients ) and the number of jobs (one job or two, or more jobs).” 

Point 5:

In the Results section, I recommend standardizing the presentation of p-values. Use either "p < 0.05" or "p ≥ 0.05" (as in Table 1), or provide the exact p-value (as in Table 2) consistently throughout.

 Response 5: In the Results section, we have standardized the presentation of p values: "p < 0.05" or "p ≥ 0.05" (as in Table 1) 

Point 6:

In the title of Table 4, it is unclear what is being predicted—is it overweight/obesity? Is binary or ordinal regression used? Please clarify how the dependent variable was constructed. Additionally, ensure the presentation of p-values is consistent.

 Response 6: Thank you for your valuable comments. We consulted a statistician. As suggested by the reviewer, we have provided other information. Table title changed: ,,The impact of eating behaviors on the occurrence of overweight/obesity (BMI ≥25 kg/m2)- a binary regression”. Added in section Materials and Methods: ,,Additionally, binary logistic regression was used to predict the presence of overweight/obesity based on dietary habits. The dependent variable (BMI group) was dichotomous (overweight/obese; not overweight/obese)”. The p-value presentation has been unified.    

Point 7:

Discussion, line 369: Please avoid causal language. To assess development, a prospective cohort study would be needed. This is a wrong interpretation of the cross-sectional study results. Possibly overweight subjects demonstrate lower odds of unhealthy dietary habits as a consequence of the outcome. Also, I recommend revising similar instances throughout the manuscript. You can only declare the differences in prevalence or increased odds (not risk).

 Response 7: We have made corrections in accordance with the reviewer's suggestions.  

Point 8:

Conclusion, line 388: Should this refer to incidence or prevalence? Line 393: You mention correlating, but no correlations were presented in the results. Possible demonstrating an increased odds of unhealthy dietary habits? Please revise this for accuracy.

 Response 8: We have made corrections in accordance with the reviewer's suggestions. We added: ,,In our study, overweight individuals had lower odds of unhealthy eating habits compared to obese participants. It is possible that overweight subjects demonstrated lower odds of unhealthy dietary habits as a consequence of the outcome.” 

Point 9:

Moderate English editing is required after revision with specific attention to epidemiological terms (incidence vs. prevalence, odds vs. risk, etc.).

Response 9:  English language editing has been done, with specific attention to epidemiological terms (incidence versus prevalence, opportunity versus risk, etc.)  

Reviewer 3 Report

Comments and Suggestions for Authors

Thank you for your contribution.

This study is a very informative to the primary doctors and nurses, however, I want to give you a some minor comments.

1. Is there income data in this survey? The different of income may affect to food choice as well as other factors.

2. When using logistic regression analysis, do you use  some covariates adjustment? such as income, age, BMI, etc.

3. This study results are very informative to the primary health care physicians, doctors, and nurses, however, is there any result to educate of bad diet habits by a bad diet habits of primary healthcare people in this study?

Author Response

Response to Reviewer 3 Comments

We would like to thank Reviewer 3 for the positive opinion of our manuscript.

We have carefully considered all reviewer's considerations of the paper. Please find enclosed our detailed answers to reviewer queries.

Point 1:

Is there income data in this survey? The different of income may affect to food choice as well as other factors.

Response 1:

The questionnaire did not include questions about income. 

Point 2:

When using logistic regression analysis, do you use some covariates adjustment? such as income, age, BMI, etc.

Response 2:

The independent variables age and BMI were included in the logistic regression analysis. We calculated their impact on eating/health behaviors separately.Table 4 uses BMI as the dependent variable.  

Point 3:

This study results are very informative to the primary health care physicians, doctors, and nurses, however, is there any result to educate of bad diet habits by a bad diet habits of primary healthcare people in this study?

Response 3:

Changing unhealthy eating habits is a long-term process that needs to be verified and repeated. Primary care doctors and nurses should be role models for their patients. That is why it is so important that they adopt and maintain good eating habits. Healthcare workers will be better public health educators and will motivate their patients to adopt good eating habits if they practice them themselves. Showing medical staff that they have bad eating habits may influence them to change their lifestyle.

Round 2

Reviewer 1 Report

Comments and Suggestions for Authors

Although the authors have made minor changes, the limitation of a substantially dated study still remains.

Comments on the Quality of English Language

English language seems to be OK.

Reviewer 2 Report

Comments and Suggestions for Authors

Dear Authors,

Thank you for all corrections made, good work!